# Adaptive clinical trials in surgery: A scoping review of methodological and reporting quality

Phillip Staibano[1,2]*, Emily Oulousian[1,3], Tyler McKechnie[2,4], Alex Thabane[2], Samuel Luo[1,5], Michael K. Gupta[1], Han Zhang[1], Jesse D. Pasternak[6], Michael Au[1], Sameer Parpia[2], J. E. M. (Ted) Young[1], Mohit Bhandari[2,7]

1 Division of Otolaryngology–Head and Neck Surgery, Department of Surgery, McMaster University, Hamilton, Ontario, Canada, 2 Department of Health Research Methodology, Evidence, and Impact, McMaster University, Hamilton, Ontario, Canada, 3 McGill University School of Medicine, McGill University, Montreal, Quebec, Canada, 4 Division of General Surgery, Department of Surgery, McMaster University, Hamilton, Ontario, Canada, 5 Michael G. DeGroote School of Medicine, Hamilton, Ontario, Canada, 6 Endocrine Surgery Section Head, Division of General Surgery, Department of Surgery, University Health Network, University of Toronto, Toronto, Ontario, Canada, 7 Division of Orthopedic Surgery, Department of Surgery, McMaster University, Hamilton, Ontario, Canada

* phillip.staibano@medportal.ca

**Data Availability Statement:** All relevant data are available on OSF: 10.17605/OSF.IO/2GDC5.

**Funding:** The author(s) received no specific funding for this work.

## Abstract

### Importance

Adaptive surgical trials are scarce, but adopting these methods may help elevate the quality of surgical research when large-scale RCTs are impractical.

### Objective

Randomized-controlled trials (RCTs) are the gold standard for evidence-based healthcare. Despite an increase in the number of RCTs, the number of surgical trials remains unchanged. Adaptive clinical trials can streamline trial design and time to trial reporting. The advantages identified for ACTs may help to improve the quality of future surgical trials. We present a scoping review of the methodological and reporting quality of adaptive surgical trials.

### Evidence review

We performed a search of Ovid, Web of Science, and Cochrane Collaboration for all adaptive surgical RCTs performed from database inception to October 12, 2023. We included any published trials that had at least one surgical arm. All review and abstraction were performed in duplicate. Risk of bias (RoB) was assessed using the RoB 2.0 instrument and reporting quality was evaluated using CONSORT ACE 2020. All results were analyzed using descriptive methods.

**Competing interests:** The authors have declared that no competing interests exist.

## Findings

Of the 1338 studies identified, six trials met inclusion criteria. Trials were performed in cardiothoracic, oral, orthopedic, and urological surgery. The most common type of adaptive trial was group sequential design with pre-specified interim analyses planned for efficacy, futility, and/or sample size re-estimation. Two trials did use statistical simulations. Our risk of bias evaluation identified a high risk of bias in 50% of included trials. Reporting quality was heterogeneous regarding trial design and outcome assessment and details in relation to randomization and blinding concealment.

## Conclusion and relevance

Surgical trialists should consider implementing adaptive components to help improve patient recruitment and reduce trial duration. Reporting of future adaptive trials must adhere to existing CONSORT ACE 2020 guidelines. Future research is needed to optimize standardization of adaptive methods across medicine and surgery.

## Introduction

Randomized-controlled trials (RCTs) are essential for evaluating the effectiveness and safety of interventions in healthcare [1]. Their importance is reflected in the literature: since 1965, over 39,000 RCTs have been published globally, with over 60% published in the last 20 years [2]. In 2003, however, only 3.4% of studies published in leading journals were surgical RCTs [3]. Despite a 50% increase in the number of published surgical trials between 1999 and 2009, this number has remained stable over the past decade [4]. Surgical trials suffer from a high rate of discontinuation and nonpublication rates often due to slow patient recruitment [5]. A systematic review of surgical trials published from 2008 to 2020 highlighted several methodological concerns with surgical RCTs, including small sample sizes, a focus on minor clinical outcomes, moderate-to-high bias, and inconsistent usage of blinding and expertise-based randomization [3, 6].

In medicine, at least 50% of adopted interventions are derived from RCTs, yet fewer than 25% of surgical interventions are based on evidence derived from RCTs [3]. Adherence to clinical trial methodological standards in surgery is often impacted by high costs, feasibility issues, between-group crossover, and poor patient adherence [7, 8]. As a consequence, many surgical innovations have been adopted based upon non-scientific practices and small-scale, poorly controlled observational studies [6]. Randomized studies in surgery, however, have historically led to an effective discarding of unnecessary surgical procedures [9].

The conventional randomized trial design with a large sample size remains the gold-standard approach for comparing medical interventions. Classically, RCTs adhere to a fixed study protocol and culminate in a pre-defined final analysis. Adaptive clinical trials, however, involve flexible adjustments to the study protocol based upon pre-specified interim analyses, which can permit sample size re-calculations, adding or dropping treatment arms, and/or stopping the trial for futility or lack of efficacy (Table 1). Adaptive trials are gaining popularity in drug development and other medical disciplines, as demonstrated by the TAILoR, 18-F PET, and STAMPEDE trials [10–12]. Benefits of trial adaptability include cost reduction, decreased probability of assigning patients to an ineffective treatment arm, and expedited trial completion [13]. Adaptive trials were useful during the peak of COVID-19 to accelerate the

**Table 1. Advantages and disadvantages of adaptive trial designs.**

| Advantages | Disadvantages |
| --- | --- |
| • Early study termination or dropping treatment arm due to efficacy, futility, or harm.<br>• Improve statistical power.<br>• Maximize probability that patient is recruited to better-performing treatment group.<br>• Reduce costs associated with trial design and implementation.<br>• Reduce the sample size.<br>• Reduce the number of trials needed to address a clinical question.<br>• Reduce time to trial reporting. | • Difficult to secure funding.<br>• Less applicable if delayed time to outcome assessment.<br>• Risk of introducing bias if not well-designed<br>• Statistically onerous<br>• Requires reliable research infrastructure. |

comparison of anti-viral therapies [14]. Adaptive methodology, however, has been less adopted in evidence-based surgery. Given the challenges surgical researchers face in implementing conventional RCTs, adaptive trials may represent a high-quality alternative that allows surgical researchers to retain the benefits of randomization whilst minimizing costs and permitting protocol adjustments to maximize trial feasibility, adherence, and validity. The goal of this study was to perform a scoping review to characterize the methodological and reporting quality of adaptive study designs in surgical trials. These findings will define the current landscape of adaptive surgical trial quality so that they can be optimized and applied to future surgical populations.

## Materials and methods

### Study design

We performed a scoping review of all prospective randomized trials that have employed adaptive designs within any surgical discipline. Surgical disciplines for the purposes of this study included cardiothoracic surgery, general surgery, gynecological surgery, orthopedic surgery, ophthalmology, otolaryngology, neurosurgery, plastic surgery, and urological surgery. We included all studies that compared at least one surgical arm. We also included studies that compared at least one surgical arm to a non-surgical interventional procedure. We registered this review in Open Science Framework (DOI: 10.17605/OSF.IO/2GDC5). Due to the nature of this study, institutional review board (IRB) approval was not required. This review was performed in accordance with PRISMA Scoping Review guidelines [15].

### Search strategy

We performed a library citation search of Medline (Ovid), EMBASE (Ovid), Web of Science, Cochrane Collaboration, and CENTRAL databases from inception to October 14th, 2023 (S1 File). The search strategy was finalized by a librarian specialist. We performed a search of pre-print databases to identify any relevant drafted manuscripts or ongoing clinical trials. We also performed a search of clinicaltrials.gov using the keywords "adaptive/Bayesian", "clinical trial/trial", and "surgery/surgical" to evaluate for any active surgical trials meeting our eligibility criteria. We also used keywords for each included surgical discipline. We screened the first 10 pages of relevant results of clinicaltrials.gov and pre-publication databases. Additional studies were identified through reference list searches of included articles. All duplicates were removed, and citations were managed using Covidence software (Melbourne, Victoria, Australia) [16].

### Study selection

Any adaptive trial investigating one or more surgical or procedural interventions was included. Eligible trials must have included one of the following adaptive designs: Bayesian, frequentist, sample re-estimation, group sequential, multi-arm multi-stage (MAMS), seamless, continual reassessment, population enrichment, adaptive randomization, and/or adaptive dose-ranging. Any protocols for adaptive surgical trials were collated for the discussion, but not included in the final article synthesis. We excluded any trials that did not include at least one surgical arm, as well as any trials evaluating perioperative medications or non-surgical interventions provided in a surgical setting. Abstracts and conference proceedings, non-human studies, and non-English publications were also excluded.

### Outcomes of interest and data abstraction

The primary outcome of this study was to characterize the existing adaptive surgical trial literature and assess the methodological and reporting quality. All identified citations underwent screening of titles and abstracts in duplicate (E.O. and S.L.), followed by full-text evaluation in duplicate (P.S. and E.O.). With the use of a standardised and piloted data abstraction template, the following study characteristics were extracted: publication year, country of study, study design, and methodological details pertaining to adaptive design. We used CONSORT ACE 2020 to guide data abstraction and identified the following adaptive trial characteristics: type of adaptive design, number and type of pre-determined interim analyses, goals of interim analysis, presence of any statistical simulations, and details related to randomization, blinding, type I error adjustments, and final statistical analysis [17]. All data abstraction was performed in duplicate (E.O. and S.L.) and any conflicts resolved by third reviewer (P.S.).

### Quality appraisal

For all studies meeting eligibility criteria, we performed quality appraisal using the Cochrane Risk-of-Bias for Randomized Trials (RoB 2.0) instrument [18]. We also performed a reporting quality appraisal using the CONSORT ACE 2020 guidelines for adaptive trials [17]. The author checklist was applied to the abstract and main text for all included articles. We categorized quality of reporting into *fully reported*, *partially reported (with details provided*, *where relevant)*, and *not reported*. All quality appraisal was performed in duplicate (E.O. and S.L.). All conflicts were resolved via discussion and a third reviewer (P.S.).

### Statistical analysis

We performed descriptive statistical analysis for all included studies. We reported all continuous outcomes as means (with standard deviation) or median (with ranges), where applicable. All categorical outcomes were reported as proportions and percentages, where applicable. All analyses were performed in Microsoft Excel (Redmond, Washington, USA).

## Results

### Search strategy and article selection

Our database search yielded 1338 results from database inception to October 2023, of which, six published trials met eligibility criteria (i.e., <0.5% of retrieved citations) (Fig 1) [19–24]. Our review of clinicaltrials.gov yielded 263 active trials, but none met the eligibility criteria. Our review of pre-publication databases did not yield any manuscripts that met eligibility criteria.

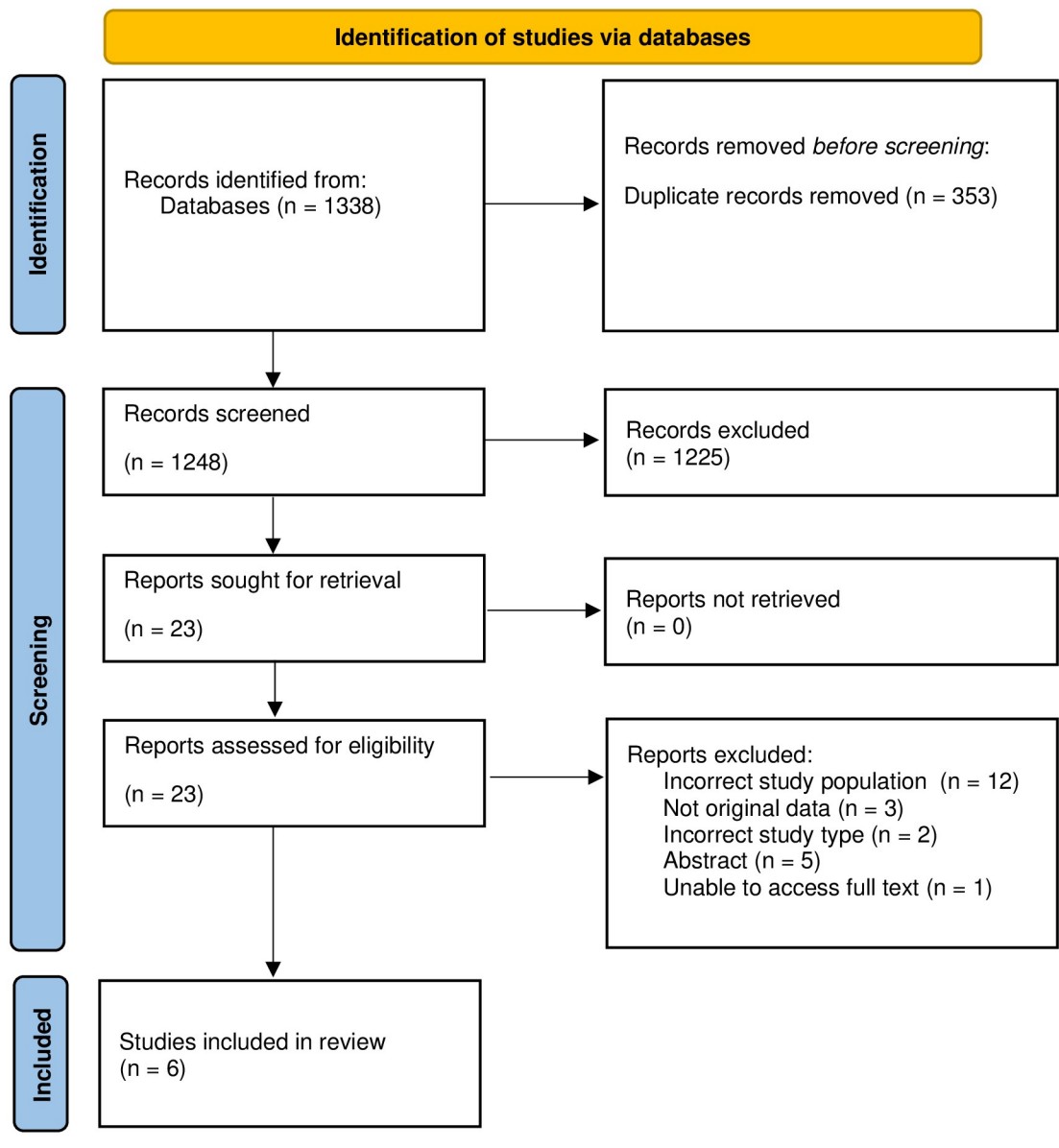

**Fig 1. PRISMA flow chart for study inclusion.**

## Study characteristics

All six included studies were published after 2015 (Table 2) [19–24]. They were in the fields of cardiothoracic surgery, orthopedics, urology, and oral and maxillofacial surgery. Five studies published their study protocol in advance of trial registration [19–23]. All studies compared two treatment arms. Two trials were conducted in the US (33.3%), one in Japan (16.7%), and three in Europe (50%). Two trials were multi-institutional [21, 23]. The total number of recruited patients was 2980 and the range of patient recruitment was 30–1746 patients. We also note that three trials recruited under 150 patients [20–22]. All studies utilized at least one surgical treatment arm. One cardiothoracic surgery trial utilized an interventional procedure as an experimental arm [23]. All studies utilized 1:1 randomization, and blinding included a

**Table 2. Summary of included studies.**

| Author (year) | Study design | Country | Surgical field | No. of centres | No. patients recruited | No. patients analysed | Treatment groups — Experimental | Treatment groups — Control | Primary outcome | Secondary outcomes | Type of adaptive design | Blinding | Data analysis plan | Type of randomization | Study follow-up | Trial adjustments made at interim analysis | Published protocol |
|---|---|---|---|---|---|---|---|---|---|---|---|---|---|---|---|---|---|
| Gaudino et al. (2021) [19] | Prospective RCT (superiority) | USA | Cardiothoracic surgery | 1 | 420 | 420 | Posterior left pericardiotomy (n = 212) | No intervention (n = 208) | Rate of postoperative atrial fibrillation during hospital admission | Cumulative time spent in atrial fibrillation, need for antiarrhythmic medications and systemic anticoagulation, the need for cardioversion, hospital readmission, and the duration of hospital admission | Group sequential; sample size re-estimation; two pre-specified efficacy interim analyses | Patients and outcome assessors | ITT; sensitivity analysis as per as-treated principle | 1:1… | 1 month | Sample size increased at second interim analysis due to lower-than-expected rate of primary outcome | Yes |
| Mastroianni et al. (2022) [20] | Prospective RCT (superiority) | Italy | Urology | 1 | 116 | 116 | Robot-assisted radical cystectomy (n = 58) | Open radical cystectomy (n = 58) | Perioperative transfusion rate reduction | Perioperative outcomes during hospital stay, complication rates, global cost analysis, and 6-month functional, oncologic and HRQoL outcomes | Covariate adaptive randomization | Outcome assessors (i.e., pathologists who assessed surgical specimens) | None reported | | 6 months | None reported | Yes |
| Metcalfe et al. (2022) [21] | Prospective RCT (superiority) | UK | Orthopedic surgery | 24 | 117 | 114 | Debridement with device (In-Space® balloon) (n = 56) | Debridement only (n = 61) | Oxford Shoulder Score at 12 months postoperatively | Constant Score, range of pain-free flexion/abduction, Western Ontario Rotator Cuff index, EuroQol EQ-5D-5L, change in symptoms, participant global impression of change, resource use, and adverse events | Group sequential; sample size re-estimation; two pre-specified interim analyses for futility and efficacy and futility, respectively | Patients (until 12 months after surgery and outcome assessors | mITT | | 12 months | Recruitment and randomisation stopped at the first interim analysis based on futility criteria | Yes |
| Neuberger et al. (2023) [22] | Prospective RCT (superiority) | Germany | Urology | 1 | 551 | 544 | TP-RARP with peritoneal flap (n = 270) | TP-RARP without peritoneal flap (n = 274) | Rate of symptomatic lymphoceles needing surgical intervention | Asymptomatic lymphoceles, perioperative parameters (i.e. length of hospital stay), and postoperative complications | Group sequential; sample size re-estimation; one pre-specified interim analysis for futility and efficacy | Surgeons blinded intraoperatively until end of surgery, patients and outcome assessors were blinded for the entirety of the trial | ITT; per-protocol and as-treated for primary outcome | | 6 months | Sample size increased at interim analysis | Yes |
| Reardon et al. (2017) [23] | Prospective RCT (non-inferiority) | Canada, Europe, USA | Cardiothoracic surgery | 87 | 1746 | 1,660 | TAVR (n = 879) | Surgical AVR (n = 867) | Composite of death from any cause or disabling stroke at 24 months | Major adverse cardiovascular and cerebrovascular events, myocardial infarction, all types of strokes, any surgical reintervention | Bayesian design and analysis | Not reported | mITT; sensitivity analysis performed for patients lost to follow up | | 24 months | None reported | Yes |
| Yoshioka et al. (2018) [24] | Prospective RCT (superiority) | Japan | Oral and Maxillofacial Surgery | 1 | 30 | 30 | Intraoral vertical ramus osteotomy (n = 15) | Sagittal split ramus osteotomy (n = 15) | Cephalogram measurements of proximal and distal osteotomy segments | None reported | Adaptive randomization | NR | None reported | NR | 12 months | None reported | No |

HRQoL, health-related quality of life; ITT, intention-to-treat; mITT, modified intention-to-treat; NR, not reported; RCT, randomized clinical trial; TAVR, trancatheter aortic valve replacement; TP-RARP, Transperitoneal robotic assisted radical prostatectomy

combination of patients, outcome assessors, and/or surgeons. These studies did not report an expertise-based group allocation design. Primary outcomes in five trials clinical endpoints of direct patient relevance [19–23]. In the five studies that reported secondary outcomes, the majority were further clinical endpoints with two studies evaluating patient-centred outcomes, and one performing a global economic analysis. The study follow-up range for all trials was 1–24 months with most studies using a 6–12-month study follow-up period. All study data is included in S2 File.

## Adaptive methodology characteristics

All included studies were prospective adaptive RCTs (Table 3). One study specified a non-inferiority design [23]. Two studies employed adaptive randomization [20, 24]. Mastroianni et al. (2021) utilized covariate adaptive randomization and identified their pre-specified prognostic strata in their methods [20]. Yoshioka et al. (2018) did not describe their adaptive randomization methods [24]. Metcalfe et al. (2022) and Reardon et al. (2017) mentioned the role of statistical simulations in informing their trial design and analysis plan [21, 23]. Four studies described their proposed statistical analysis plan, including the type of group allocation analysis and the relevant inferential testing methods. Reardon et al. (2017) was the only study to employ a Bayesian method for trial design and analysis, and this group also used a statistical consulting firm to assist with calculations [23]. Three studies described a group-sequential design with two treatment arms alongside sample size re-estimation performed at each interim analysis. Four studies also described their adaptive methodologies, including the number and goal(s) of their interim analyses [19, 21–23]. These interim analyses were all pre-specified and performed for the purpose of study termination for futility or efficacy, and/or sample size re-estimation. In two studies, the sample size was increased at the interim analysis [19, 22]. Two studies utilized two pre-specified interim analyses [19, 21]. Gaudino et al. (2021) reported a sample size increase at the second interim analysis due to a lower-than-expected primary outcome event rate [19]. Metcalfe et al. (2022) stopped further randomization due to futility criteria being met [21].

## Risk of bias and CONSORT reporting

All trials were evaluated by two independent reviewers in duplicate using the Cochrane RoB 2.0 instrument (Fig 2). Overall, three (50%) trials had a low risk of bias and three (50%) had a high risk of bias. A high risk of bias was primarily derived from the randomization process, deviations from intended interventions, and the reporting of outcomes [20, 23, 24]. We also performed an assessment of reporting quality using the CONSORT extension ACE checklist [17]. Two studies directly referenced CONSORT ACE 2020 guidelines [20, 22]. Most studies (83.3%) reported trial registration, protocol, full statistical analysis plan, and funding sources. We described abstract and main text reporting outcomes for all included studies (S1 Table; S3 File). Despite other studies adequately reporting details for rationale, methods, and results, one study reported adaptive design details within their abstract [21]. We found that within the methods section of the main text, most reporting heterogeneity (i.e., 50–83.3% of studies describing either partial or no reporting) occurred when describing design changes following trial initiation, changes to study design after the start of the trial, and details regarding blinding implementation and adherence (Fig 3). In the results section, no studies reported the reasons for trial stoppage or sufficient details surrounding this decision, and there was heterogeneity in the quality of outcome reporting. Lastly, regarding the discussion, we found that findings were adequately contextualized, but only two studies fully reported study limitations with direct reference to adaptive design decisions [21, 22].

**Table 3. Adaptive design characteristics for included studies.**

| Author (year) | Adaptive design | Prespecified interim analysis | Goals of interim analysis | Change(s) to design after start of trial | Statistical simulations | Goals of simulation | Modelling algorithm | Sample size calculation | Type of randomization | Blinding | Type of statistical analysis | Type I error adjustments | Adaptive design reporting | Statistical resources |
|---|---|---|---|---|---|---|---|---|---|---|---|---|---|---|
| Gaudino et al. (2021) [19] | Group sequential; sample size re-estimation | 1. Sample-size re-estimation 2. Two pre-specified blinded efficacy interim analyses 3. No futility interim analysis | Sample size re-estimation | Second interim analysis: Primary outcome event rate lower than anticipated and so, trial sample size was increased | None reported | NR | Not reported | Yes, 90% power to detect 50% reduction in the primary outcome; based on previous studies | 1:1; mixed-block randomisation sequence | 1. Patients and outcome assessors unblinded after study completion 2. All parties unmasking if complications secondary to treatment group | 1. ITT for primary outcome 2. Sensitivity analysis for primary outcome per as-treated principle 3. ITT for secondary outcomes 4. As-treated principle for safety outcomes | Type I error rate of 0.05 (two-sided); no adjustments for biases at interim analysis | Yes, mentioned in title, methods, and results | All analyses performed using R software |
| Mastroianni et al. (2022) [20] | Covariate-adaptive randomization | None specified | NR | None | No | NR | Not reported | Yes, 80% power to detect 5% difference in primary outcome; based on previous studies | 1:1 based upon BMI, ASA score, baseline hemoglobin, planned urinary diversion, neoadjuvant chemotherapy, and clinical tumour stage | 1. Outcome assessors (i.e., pathologists) | Details not reported | Type I error rate of 0.05; no further details reported | Yes, mentioned in methods | All analyses performed using SPSS |
| Metcalfe et al. (2022) [21] | Group sequential; sample size re-estimation; covariate-adaptive randomization | 1. Sample size re-estimation 2. Two pre-specified interim analyses: 3. First interim analysis done for futility 4. Second interim analysis done for futility and efficacy | Sample size re-estimation | Recruitment and randomisation stopped after futility boundary was crossed at the first interim analysis | Yes, performed at start of trial | To determine predefined interim stopping threshold | Not reported | Yes, 90% power to detect a 6-point MCID with 15% loss to follow-up rate; based on previous studies | 1:1 based upon site, sex, age, and cuff tear size | 1. Patients (blinded for 12 months) and outcome assessors | 1. mITT for primary outcome (adjusted for interim analysis) 2. Low rate of missing data so did not impute missing data points | Type I error rate of 0.05 (two-sided); adjustment for bias planned if study progressed beyond first interim analysis | Yes, mentioned in title, methods, and results | All analyses performed using R software |
| Neuberger et al. (2023) [22] | Group sequential; sample size re-estimation | 1. Sample size re-estimation 2. One pre-specified interim analysis for futility and efficacy | 1. Trial termination for futility or efficacy 2. Sample size re-estimation | Primary interim analysis: Sample size was increased | No | NR | NR | Yes, based on previous studies | 1:1 | 1. Surgeons unblinded after surgery 2. Patients and assessors blinded until study completion | Yes | Type I error rate of 0.025 (one sided); no adjustments for biases at interim analysis | Yes, mentioned in methods and results | All analyses performed using JMP and R software |
| Reardon et al. (2017) [23] | Bayesian design and analysis | 1. Bayesian analytical methods for non-inferiority 2. Bayesian interim analysis when 1400 patients reached 12-month follow-up | Sample size re-estimation | Primary interim analysis: No stated methodological changes | Yes, performed at start of trial | To determine Bayesian posterior probability for non-inferiority | Yes | Yes, performed using Bayesian analysis and based upon 17% incidence rate of primary outcome | 1:1 stratified by site and need for revascularization | Open label | 1. mITT for primary and secondary outcomes 2. Bayesian analogues of frequentist tests and posterior probabilities | Type I error rate of 0.05 (two-sided) used to calculate relevant Bayesian posterior probability for hypothesis testing | Yes, mentioned in methods and results | All analyses performed in conjunction with independent statistical group |
| Yoshioka et al. (2018) [24] | Adaptive random assignment procedure | NR | NR | NR | NR | NR | NR | NR | NR | NR | NR | No | Yes, mentioned in methods | Not specified |

ITT, intention-to-treat; mITT, modified intention-to-treat; NA, not applicable; NR, not reported

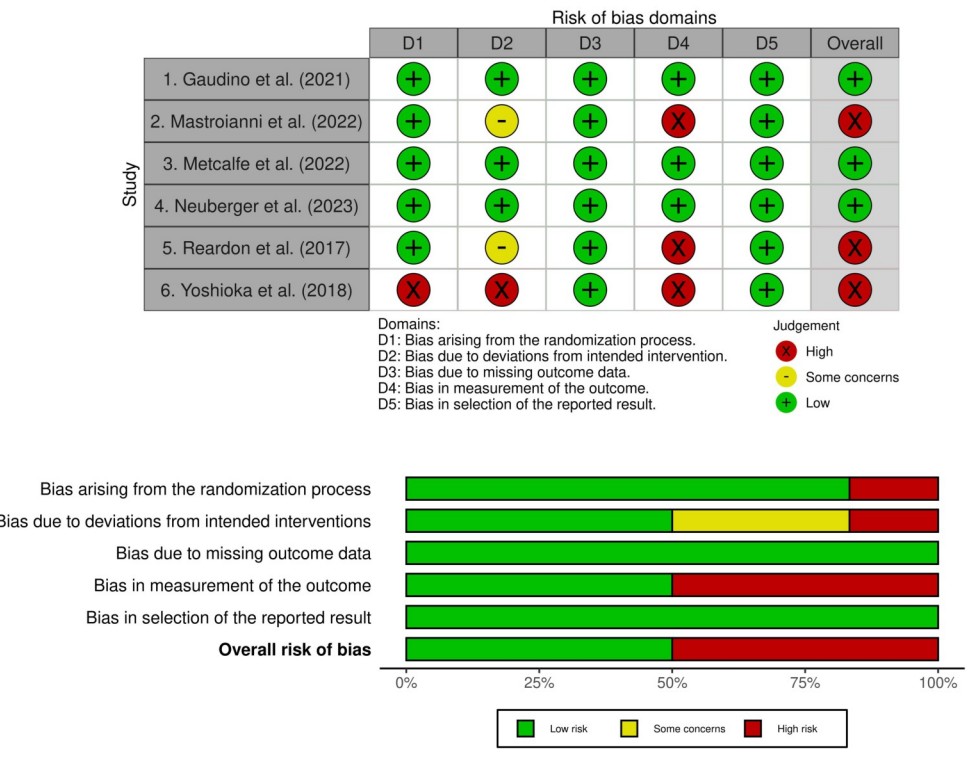

Fig 2. Risk of bias assessment for included studies.

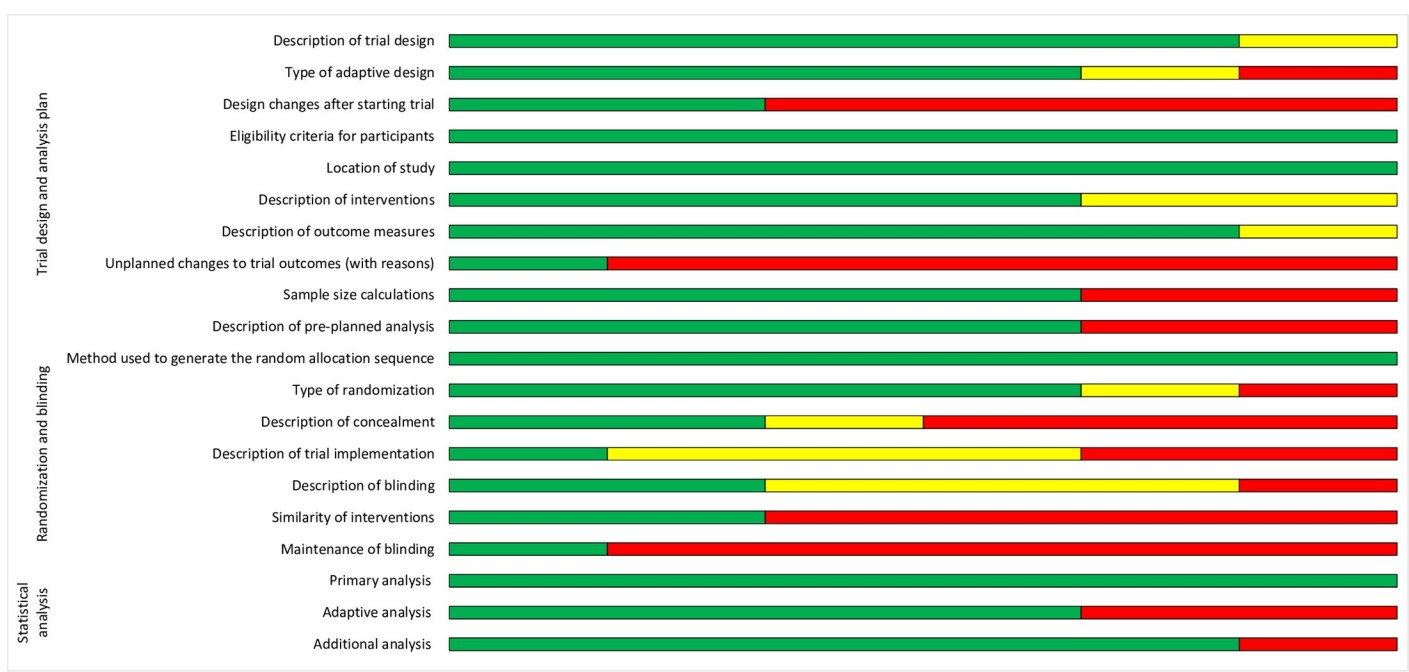

Fig 3. CONSORT ACE 2020 assessment of methodology reporting for included studies. Green = Fully reported; Yellow = Partially reported; Red = Not reported.

## Discussion

We present the first scoping review of adaptive clinical trials in surgery. Since 2015, six adaptive trials have been published comparing at least one surgical intervention, which only represents 0.5% of the database search. We also note that three trials recruited under 150 patients. Adaptive methodologies are increasingly being adopted in non-surgical trials, but their design remains nebulous and as such, they have seldom been applied to surgical populations [13].

Adaptive randomization methods represent an early adaptive method applied to trial design in medicine [25, 26]. Response-adaptive randomization can minimize the number of patients randomized to an inferior treatment arm and may suit multi-arm trials or trials with prolonged recruitment and moderately delayed responses,[27, 28]. This method of randomization may improve surgical trial recruitment and ethicality, as patients will have an increased probability of being randomized to an effective surgical arm [28]. Sirkis et al. (2022) simulated adaptive randomization techniques in the RECOVERY trial demonstrating that adaptive approaches may have reduced death and increased the likelihood of randomization to effective COVID-19 therapies [29]. Adaptive randomization has the potential, however, to optimize patient recruitment in surgical trials, but must be used prudently, as surgical trials are often small-to-moderate in size and therefore prone to prognostic imbalances. *A priori* trial simulation studies to guide sample size calculations and evaluate optimal adaptive randomization methods and their biases may further inform adaptive designs for surgical trials.

Outside of adaptive randomization alone, we found that three studies employed a group sequential design with blinded sample size re-estimation performed at one or two prespecified interim analyses [19, 21, 22]. A recent systematic review identified 27 adaptive trials that utilized sample size re-estimation, in particular, this design was used in Phase I/II trials wherein there was insufficient prior knowledge of treatment group [30]. Sample size re-estimation, however, can lead to an inflated type I error rate, and though methods for optimizing these statistical adaptations exist, they remain poorly standardized within the adaptive trial literature [31]. Group sequential trial designs are defined by at least one interim analysis being built into the trial design to evaluate efficacy and/or futility of treatment arms and can facilitate the adding or dropping of additional treatment arms [32]. Sequential designs have been used in pharmaceutical trials for decades, and more recently were employed in the DEVELOP-UK trial evaluating lung perfusion following lung transplantation [32, 33]. Two studies in this review reported that statistical simulations were performed to guide design and interim analyses [21, 23]. Statistical simulations represent another advantage of adaptive trials, as they can estimate efficiency gains and facilitate trial improvements prior to funding expenditure and beginning patient recruitment [34]. Reardon et al. (2017) utilized a Bayesian analysis plan to design and perform analysis within their trial [23]. Though statistically more complicated, Bayesian-designed trials hold promise in comparative effectiveness trials wherein prior population knowledge is lacking, as these designs can assist sample size estimations, improve power, and potentially increase the rate at which patients are randomized to effective treatments [35]. Current barriers to adaptive trial adoption amongst stakeholders, however, include patient consent, risks for bias, type I error rate, lack of clear rationale, and a paucity of education about adaptive methodologies [36].

Interestingly, combining adaptive methods such as sample size re-estimation couched within a response-adaptive RCT, may also prove helpful as the synergized application of these methods can further improve power and shorten trial duration [37]. Reardon et al. (2017) reported that an independent statistical group assisted with their analysis [23]. Better adoption of complex adaptive trial methodologies amongst surgeons will need to occur alongside improved biostatistical training for surgeons and increased collaboration with statisticians [38, 39]. Moreover, it is

not difficult to conceptualize the interaction that can exist between generative artificial intelligence and the computations that underly adaptive trial design and execution [40].

We performed a quality appraisal using the Cochrane Rob 2.0 instrument for RCTs [18]. We were unable to identify a quality appraisal tool tailored to adaptive trial methodologies, but our research group is currently in the process of creating a customized adaptive trial RoB instrument. Our quality appraisal identified 50% of the included trials to have a high risk of bias, which was primarily derived from randomization details and inadequate outcome reporting. We used the CONSORT ACE 2020 guidelines to assess reporting quality in all included trials [17]. Within the main text, heterogenous reporting was primarily identified in the description of trial design, changes following the start of the trial, and details regarding blinding and concealment maintenance. These deficiencies are important to highlight, as adherence to pre-specified trial design parameters and interim analyses are critical in minimizing type I error rate and bias within adaptive methodologies. It will be important for future adaptive trials to adhere to the published CONSORT ACE 2020 guidelines. Our study limitations are the exclusion of any abstracts or grey literature, and despite liberal search criteria, the risk of missing any relevant surgical trials.

Surgical subspecialties are beginning to explore the role for adaptive trials in their respective disciplines [41, 42]. As we peer into the future of evidence-based surgery, we must identify the existing barriers to adoption and global dissemination of surgical trial design and implementation. For instance, our group is currently exploring the importance of pilot trials in surgery to overcome issues of early trial termination [5, 43]. As researchers characterize the importance of creativity in surgical innovation, we posit that adaptive trial methodologies provide yet another methodological tool to answer translational questions and advance evidence-based surgical knowledge [44]. Since conventional RCTs are often conducted out of high-income countries, cost-effective and efficient adaptive methodologies may facilitate practice-changing trials being conducted in low-middle income countries, thereby improving global collaboration and conclusion generalizability [45].

Adaptive designs have the potential to optimize patient recruitment and statistical power when sample sizes are small, or little is known about the research populations under investigation: common issues encountered in surgical trials [46]. These designs may also assist with conducting trials in rare disease populations [47]. Like any technological advancement, however, it is imperative that we always identify the appropriate research question and clinical scenario where adaptive methodologies can be best deployed, as they are not without their shortcomings [48]. For instance, adaptive trials are likely not best suited when there is a notable delay in outcome assessment or there is inadequate trial infrastructure at the primary institution. An adaptive design is, however, likely best suited when the gain in trial efficiency and cost-effectiveness greatly outweighs the added complexity to trial methodology and statistical analysis [48]. Here, we demonstrate that the number of published adaptive surgical trials is low and reporting of complex adaptive methods is often heterogenous and inadequate. Future adaptive trials must be reported in accordance with published CONSORT ACE 2020 guidelines [17]. As these methodologies continue to be optimized, we suggest that surgical trialists consider implementing adaptive design components when deemed appropriate for their clinical question and population-of-interest. Adaptive trial designs may help to improve the quality of surgical evidence, streamline time to reporting, and compliment the accelerated pace of innovation in surgery.

## Supporting information

**S1 File. Ovid search algorithm.**
(DOCX)

**S2 File. Raw data for all included trials.**
(XLSX)

**S3 File. Raw data for CONSORT ACE 2020 evaluation.**
(XLSX)

**S1 Table. CONSORT ACE 2020 assessment for included studies.**
(DOCX)

**S1 Checklist. Preferred Reporting Items for Systematic reviews and Meta-Analyses extension for Scoping Reviews (PRISMA-ScR) checklist.**
(DOCX)

## Author Contributions

**Conceptualization:** Phillip Staibano, Emily Oulousian, Tyler McKechnie, Alex Thabane, Samuel Luo, Michael K. Gupta, Han Zhang, Jesse D. Pasternak, Michael Au, J. E. M. (Ted) Young, Mohit Bhandari.

**Data curation:** Phillip Staibano, Emily Oulousian, Tyler McKechnie, Alex Thabane, Samuel Luo, Michael K. Gupta, Han Zhang, Jesse D. Pasternak, J. E. M. (Ted) Young, Mohit Bhandari.

**Formal analysis:** Phillip Staibano, Emily Oulousian, Tyler McKechnie, Alex Thabane, Samuel Luo, Michael K. Gupta, Han Zhang, Jesse D. Pasternak, J. E. M. (Ted) Young.

**Funding acquisition:** Phillip Staibano, Emily Oulousian, Michael K. Gupta, J. E. M. (Ted) Young.

**Investigation:** Emily Oulousian, Samuel Luo, Michael K. Gupta, Sameer Parpia, J. E. M. (Ted) Young, Mohit Bhandari.

**Methodology:** Phillip Staibano, Emily Oulousian, Samuel Luo, Sameer Parpia, J. E. M. (Ted) Young, Mohit Bhandari.

**Project administration:** Tyler McKechnie, Sameer Parpia, J. E. M. (Ted) Young, Mohit Bhandari.

**Resources:** Tyler McKechnie, J. E. M. (Ted) Young, Mohit Bhandari.

**Software:** Phillip Staibano, Sameer Parpia.

**Supervision:** Alex Thabane, Michael Au.

**Validation:** Alex Thabane, Michael Au.

**Visualization:** Alex Thabane, Han Zhang.

**Writing – original draft:** Phillip Staibano, Emily Oulousian, Tyler McKechnie, Alex Thabane, Michael K. Gupta, Han Zhang, Jesse D. Pasternak, Michael Au, Sameer Parpia, Mohit Bhandari.

**Writing – review & editing:** Phillip Staibano, Emily Oulousian, Tyler McKechnie, Alex Thabane, Samuel Luo, Michael K. Gupta, Han Zhang, Jesse D. Pasternak, Michael Au, Sameer Parpia, J. E. M. (Ted) Young, Mohit Bhandari.

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
