## [Decision Letter · Decision Letter 0]

4 Feb 2024

PONE-D-23-43710Adaptive clinical trials in surgery: A scoping review of methodological and reporting qualityPLOS ONE

 Dear Dr. Staibano,

Thank you for submitting your manuscript to PLOS ONE. After careful consideration, we feel that it has merit but does not fully meet PLOS ONE’s publication criteria as it currently stands. Therefore, we invite you to submit a revised version of the manuscript that addresses the points raised during the review process.

Please carefully read the reviewer's comments and answer them clearly point by point. Please also make the needed changes in the manuscript and indicate them clearly.

We look forward to receiving your revised manuscript.

Kind regards,

Frances Chung, MBBS, MD, FRCPC

Academic Editor

PLOS ONE

3. Please include your tables as part of your main manuscript and remove the individual files. Please note that supplementary tables (should remain/ be uploaded) as separate "supporting information" files

Reviewers' comments:

Reviewer's Responses to Questions

**Comments to the Author**

1. Is the manuscript technically sound, and do the data support the conclusions?

Reviewer #1: Yes

Reviewer #2: Yes

2. Has the statistical analysis been performed appropriately and rigorously? 

Reviewer #1: Yes

Reviewer #2: Yes

3. Have the authors made all data underlying the findings in their manuscript fully available?

Reviewer #1: Yes

Reviewer #2: Yes

4. Is the manuscript presented in an intelligible fashion and written in standard English?

Reviewer #1: Yes

Reviewer #2: Yes

5. Review Comments to the Author

Reviewer #1: I congratulate the authors for a well written manuscript on an important topic. The manuscript systematically addresses the application of adaptive RCTs in the surgical speciality. It describes the utility of the captive design, the current surgical literature employing this and goes on to evaluate its future application in the surgical field and recommendations on improving the methodological quality.

I have some minor suggestions as follows:

Abstract:

The conclusion mentions that “Surgical trialists should consider implementing adaptive components to help improve patient recruitment and reduce trial duration”.

Along with this line, the conclusion should also mention that the reporting of the adaptive design needs to be consistent and adherent to the existing CONSORT adaptive extension guidelines as heterogeneity in the reported trials was one of the main findings of this review.

Methods:

In the data extraction section, the authors have mentioned that methodological details pertaining to

adaptive design were collected. It may be useful to the readers to actually list all the details collected as it is the main outcome of this review.

Results:

1. Referencing of the studies needs to be consistent in all the 3 paragraphs of the results.

For example, the references of the 6 studies need to be added after the sentence “All six included studies were published after 2015 (Table 2).”

The references of the 5 studies need to be added after “Five studies published their study protocol in advance of trial registration.” Similarly, references need to be added whenever the studies are described for their characteristics at other places in this paragraph.

2. The paragraph on adaptive methodology details should be rephrased for better understanding to the readers. It should mention the results of the number of studies that described the adaptive design, pre-specified interim analysis, changes to trail design at the interim analysis in that order. At present, the information presented appears random in the paragraph.

Also, the use of standard terminology to describe an attribute would be important. The term "unplanned changes to trail design or outcomes appears at several places in the abstract, results and discussion. But it is not mentioned in the study summary table (Table 2). It is my understanding authors refer to the "changes to design after start of trial" in the table as the unplanned changes in design and outcomes. But using the same term in the table will add clarity to the readers.

3. The authors have to reference the studies they are describing in the paragraph on adaptive methodology (needs to be rephrased as above) and bias assessment as well.

Discussion:

In the second paragraph, the authors describe the covariate adaptive randomization which is altering randomization considering the prognostic factors. In the next line, they reference the studies for response adaptive randomization, which may actually be done during an interim analysis.

It is unclear to the readers if the 2 concepts are the same or different. It may be useful to describe the concepts in a bit more detail here.

Reference needs to be added after the sentence “Reardon et al. (2017) utilized a Bayesian analysis plan to design and perform analysis within their trial.” There are other places in the discussion where studies are mentioned and reference not added, that need to be looked into.

Reviewer #2: Summary: There are no issues with the approach to the topic; I like the adaptive clinical trials and their review, as well as the approach used by the authors for the summary. The manuscript scores well for the novelty, significance, interest, and impact of the finding on the current state of RCTs in the literature. The authors have done extensive hard work on literature search and summarizing the information; very good descriptive analysis, well-organized results, and nicely covered discussion. The tables are compact, the flow chart is clear, and the methods are appropriately used. Overall, it has a good methodology and approach.

Major Strengths: This topic is very interesting and adds good knowledge/information to the upcoming researchers on how to address some concerns while conducting RCTs.

Major Weaknesses: NIL.

6. PLOS authors have the option to publish the peer review history of their article (what does this mean?). If published, this will include your full peer review and any attached files.

Reviewer #1: No

Reviewer #2: **Yes: **Mahesh Nagappa

---

## [Author Response · Author response to Decision Letter 0]

5 Feb 2024

Thank you for your comments. You have improved our manuscript. We have addressed each point individually.

---

## [Editor Report · Decision Letter 1]

12 Feb 2024

Adaptive clinical trials in surgery: A scoping review of methodological and reporting quality

PONE-D-23-43710R1

Dear Dr.Staibano

We’re pleased to inform you that your manuscript has been judged scientifically suitable for publication and will be formally accepted for publication once it meets all outstanding technical requirements.

Kind regards,

Frances Chung, M.B.B.S,  MD, F.R.C.P.C

Academic Editor

PLOS ONE
---

## [Editor Report · Acceptance letter]

22 Mar 2024

PONE-D-23-43710R1 

PLOS ONE

Dear Dr. Staibano, 

I'm pleased to inform you that your manuscript has been deemed suitable for publication in PLOS ONE. Congratulations! Your manuscript is now being handed over to our production team.

Kind regards, 

on behalf of

Dr. Frances Chung 

Academic Editor

PLOS ONE